# Automatic Curation of Court Documents: Anonymizing Personal Data

**Diego Garat** * and **Dina Wonsever**

Instituto de Computación, Facultad de Ingeniería, Universidad de la República, Av. Julio Herrera y Reissig 565, Montevideo 11300, Uruguay; wonsever@fing.edu.uy

\* Correspondence: dgarat@fing.edu.uy

**Abstract:** In order to provide open access to data of public interest, it is often necessary to perform several data curation processes. In some cases, such as biological databases, curation involves quality control to ensure reliable experimental support for biological sequence data. In others, such as medical records or judicial files, publication must not interfere with the right to privacy of the persons involved. There are also interventions in the published data with the aim of generating metadata that enable a better experience of querying and navigation. In all cases, the curation process constitutes a bottleneck that slows down general access to the data, so it is of great interest to have automatic or semi-automatic curation processes. In this paper, we present a solution aimed at the automatic curation of our National Jurisprudence Database, with special focus on the process of the anonymization of personal information. The anonymization process aims to hide the names of the participants involved in a lawsuit without losing the meaning of the narrative of facts. In order to achieve this goal, we need, not only to recognize person names but also resolve co-references in order to assign the same label to all mentions of the same person. Our corpus has significant differences in the spelling of person names, so it was clear from the beginning that pre-existing tools would not be able to reach a good performance. The challenge was to find a good way of injecting specialized knowledge about person names syntax while taking profit of previous capabilities of pre-trained tools. We fine-tuned an NER analyzer and we built a clusterization algorithm to solve co-references between named entities. We present our first results, which, for both tasks, are promising: We obtained a 90.21% of F1-micro in the NER task—from a 39.99% score before retraining the same analyzer in our corpus—and a 95.95% ARI score in clustering for co-reference resolution.

**Keywords:** de-identification; transfer learning; NER; clusterization; co-reference resolution

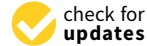



## 1. Introduction

The National Jurisprudence Base's (NJB) mission is to provide public access to the decisions of the different courts of the Judicial Branch, online and for free. Although formally jurisprudence and customs are not a source of law in our country, they constitute important precedents for all the actors in a judicial process. Since its creation in 2008, rulings from different courts have been systematically added to the NJB, including those from the Supreme Court of Justice (SCJ), the Courts of Appeals and from the Courts of First Instance (the latter still in the process of incorporation). Nowadays, the digital legal database has around 80,000 documents, publicly available on an Internet portal (http://bjn.poderjudicial.gub.uy, 12 December 2021). In Figure 1, we show an example of a document provided by the NJB portal.

Before their publication, the court rulings undergo a manual curation process. Through this process, they are enriched with information about their priority, they are classified within a legal taxonomy, they are summarized, etc. Furthermore, the very text of each court decision is modified to hide personal data considered sensitive, to prevent "the possible harmful effects of the advertising of certain data, which, if inappropriate, would cause the invasion of a protected right and the birth of conflicts, and the possible responsibility of

producers, administrators and/or distributors of any database, as it would be—or is—the Judicial Branch" [1]. In this sense, the SCJ has not only detailed the data that must be deleted by current regulations—e.g., crimes related to modesty or decency, those involving minor offenders, etc.—but has also advised against publishing data of another group of people that are not contemplated explicitly in the legislation, such as primary offenders, whistle-blowers, witnesses, etc.

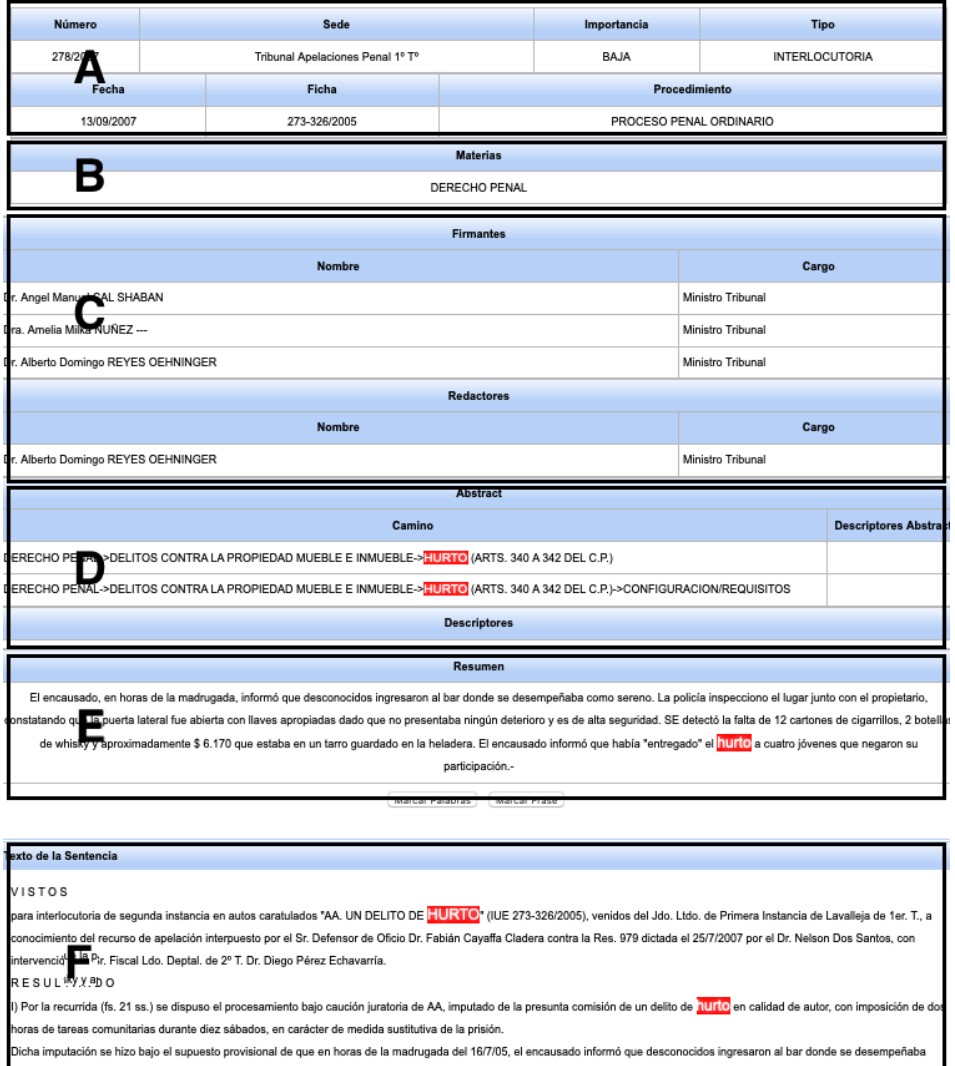

**Figure 1.** Example of a NJB decision. Published information includes: (**A**) general information of the decision: identification number, date, type of decision, etc.; (**B**) theme; (**C**) jury; (**D**) classification; (**E**) summary; and (**F**) text of the decision.

On the other hand, judicial rulings contain a large number of citations to laws, decrees, bibliographies and previous rulings, in between others, which are not recollected or systematized in any way: The reader should look for them throughout the text. They are, therefore, completely ignored during the incorporation of documents to the NJB, even though they could be useful for searching and data exploitation.

The de-identification or anonymization (Although sometimes the term 'de-identification' is used to cover a wider spectrum of entities than just anonymizing names—for example, including ages, phone numbers, addresses, etc.—in this article, we use both as synonyms) is performed manually, which makes it costly, both in time and in human resources, and without guarantees of a reliable result [2]. Most importantly, the publication of data is always delayed as a consequence of the scarcity of human resources.

The automation or semi-automation of pre-publication tasks could not only reduce the workload and publication time but also have a great impact on the user experience while interacting with the jurisprudence database. With this goal in mind, a data-driven research project was submitted to a consortium formed by the National Agency for Research and Innovation (ANII) and the Ministry of Industry and Energy. The project was presented by a research group from the School of Engineering of the Universidad de la Republica, which had signed an agreement with the Judiciary branch. Since the data before anonymization are of very restricted access, the participants in the project had to sign Non-Disclosure Agreements. An award (Project MIEM-ANII FSDA-143138) with financial support was finally obtained, and the project was then carried out, in permanent contact with specialists in the legal area linked to the Supreme Court of Justice.

Three lines of work were developed: (1) the anonymization of sensitive data, (2) the automatic classification within a legal taxonomy and (3) the detection of citations of various kinds within the texts of the judgments: to other judicial decisions, to laws and decrees or to previous arguments of referents in the field of Law.

In this paper, we focus on the de-identification of proper names, and, for this problem, there are different ways to accomplish the anonymization process. For example, the information can be completely removed, replacing it with some mark of the style "****", which is similar to crossing the information out on a written document. Let us consider the anonymization of the following fragment of a judgment of the Criminal Appeals Court (All examples of this article come from our corpus; all real names are replaced by fantasy names to preserve anonymity):

> *"That none of the officials of the agency, including the Commissioner in charge, verified or observed any kind of misplaced conduct of Mr. ****. That the complaint was made by Ms. ****, who was the head of the Prison and ordered the others (...) That the only thing Mr. **** admits is that he bought a [mobile] chip from Ms. ****, but not for that reason he can accuse you of abuse, much less rape."*

This procedure, although very effective in hiding the names of those involved, has the great disadvantage of not keeping the narrative coherent: How many actors are in the previous fragment? Are there two different men? Is the woman that sold the mobile chip the same one that made the complaint?

Therefore, our problem involves not only renaming proper names, but also assigning the same fantasy name to all references of the same entity. This task is imperative to publish the texts without revealing the true name of the actors while maintaining the coherence and understandability of the narrative of facts in the output text.

This problem might be also seen as an entity-linking task [3,4], but without an external canonical reference to attach each entity—such as, e.g., an entry of Wikipedia. It also differs in the local scope of the entities: A name is not expected to be repeated over and over again in different documents such as, for instance, a famous character over the web. Finally, given the variable number of accused, victims and witnesses involved in a trial, we don not know a priori how many different entities should be processed in each document.

In order to achieve our goal, we decided to split the solution into two tasks: firstly, recognize the named persons in the text, an NER task, and secondly, resolve the co-references between the named entities. For each choice, the performance and ease of integration are taken in consideration for future deployment into production.

For resolving the NER task, we tested several state-of-the-art NLP tools [5–8]. All of them presented issues processing our corpus due to their characteristics, which differentiates them from journalistic or scientific texts, the usual kind of text these tools are trained with. The detected problems appear in the early stages of what is a standard text analysis: The analyzers make important errors, from tokenization and segmentation to detection of named entities—particularly important for our problem. In order to tackle this issue, we perform several experiments, retraining an NER analyzer using transfer learning.

For co-reference resolution, we opted for agglomerative clustering and standard distance functions between strings. Although machine learning and clusterization have

been previously used to solve the co-reference of names and events [9–14], we opt to use a simpler solution: a matrix of string distances as input of the algorithm instead of a set of hand-crafted features or a specific algorithm. This is because, in our case, the co-reference classes only contain proper name variants.

Given the complexity of the problem and the will to obtain a working prototype, we always tried to use available tools. As we have already said, in some cases, it was not possible to use them as they were, so it was necessary to annotate and retrain. In case it is not explicitly mentioned, available tools were used, testing in general different parameter settings until a reasonable performance was achieved. This resulted in a complete prototype, comprising all stages of the problem.

The article is organized as follows. In Section 2, we report on related work, in each of the aspects we are treating. Section 3 focuses on the problem of de-identification in legal texts and why, in our case, it is important to resolve not only named entity recognition but also co-reference chains. In Section 4, we present our corpus of legal texts while in Sections 5 and 6 we present our experiments and results regarding the NER training and co-reference resolution. Finally, in Section 7, we state our conclusions and describe further work.

## 2. Related Work

The de-identification of texts has been investigated since the 1990s, especially in the domain of medical records, where confidentiality was enforced by law [15,16]. According to the relevant literature, the different approaches somehow follow the general progression of other solutions within the NLP area [17]: from the first systems in the 1990s, based solely on dictionaries, regular expressions and rules defined manually by a domain expert that obtain high precision but low coverage [2,18], going through solutions in the 2000s with Machine Learning methods such as Conditional Random Fields (CRF) or Support Vector Machines (SVM) based on attribute engineering [19–22], until the proposals of the 2010s, using Deep Neural Networks and word embeddings [23–29].

When data are in relational databases, k-anonymization algorithms are usually applied. The problem is somehow different when the information is found in free text or, even, images [17,21,30–38]. In fact, sensitive names of people, companies or hospitals might be contained within a narrative that is not always grammatically sound, increasing the difficulty of the problem. Natural Language Processing (NLP) techniques might be applied for the automatic or semi-automatic anonymization of free texts. The detection of different entities makes the process of de-identification closely related to the tasks of Recognition of Named Entities (NER) and the resolution of co-references [19,39].

For Spanish NER, several authors use CoNLL-2002 sets for evaluation: For example, in [23], an F1 score of 85.75% is reported for a CRF-LSTM model, while in [40], the F1 score goes up to 87.26% with a character+word neural network architecture. For English, training a CNN plus a fine tuning achieved a 93.5% F1 score over the CoNLL-2003 corpus [41].

Relevant references for co-reference resolution, proposing machine learning and clusterization methods are [9–14].

Specifically, the anonymization of texts from the legal domain has been explored by some recent works [42–44] reporting on some experiments in the field or just in the problems and technologies involved. In [43], a general review of the field of de-identification, suited not only for the legal domain but also for the medical domain, is presented. In [42], the authors propose to deal separately with the actual recognition of named entities and the determination of whether or not an entity should be anonymized. In our case, both features are learned together via a neural model, which seems to us to be a better option, considering that these models are especially suited for the joint learning of different aspects on the same text. On the other hand, and unlike our work, they do not make any proposal for the mapping of different textual mentions with identical reference to the same pseudonym.

Anoppi [44] is a projected service for semi-automatic pseudonymization of Finnish court judgments. Utilizing both statistics and rule-based named entity recognition methods

and morphological analysis, Anoppi should be able to pseudonymize documents written in Finnish preserving their readability and layout. However, this service was still in development by the time of the publication, and there is no detailed information about the involved processes.

It is worth mentioning that, unlike our work, neither of these papers addresses the problem of automatically assigning the same label to all references of the same entity, which is something crucial for publication in the NBJ.

### 3. De-Identification of Legal Texts

Our work focuses on the particular problem of named entity de-identification in legal texts, with the additional task of stringing of co-reference chains [39]. To exemplify the task, let us reconsider the fragment of a legal decision used as an example in the previous section:

> *Que ninguno de los funcionarios de la dependencia, incluso el Comisario a cargo constataron u observaron algún tipo de conducta fuera de lugar del Sr. **Juan Pérez**. Que la denuncia fue realizada por la Sra. **María Rodríguez** que es quien lideraba la Cárcel y ordenaba a las demás (. . . ) Que lo único admitido por el Sr. **Pérez** es que compró un chip a la Sra. **Juana Fernández**, pero no por eso se lo puede acusar de abuso y mucho menos de violación.*

> *That none of the officials of the dependency, including the Commissioner in charge, stated or observed any type of misplaced conduct by Mr. **Juan Pérez**. That the complaint was made by Mrs. **María Rodríguez** who led the jail and ordered the others (. . . ) that the only thing admitted by Mr. **Pérez** is that he bought a chip from Mrs. **Juana Fernández**, but that does not mean he can be accused of abuse and much less of rape.*

For the de-identification of named entities, the simplest solution consists of completely eliminating real names, replacing them with some generic label such as "****", similar to striking out words in a printed document. This procedure, although effective for hiding names, does not allow to distinguish between the different actors, making it difficult, if not impossible, to interpret the story correctly. As shown in the previous section, if both "María Rodríguez" and "Juana Fernández" are redacted, the reader could doubt if the lady that made the complaint is, or not, actually the same as the one that sold the phone chip.

To avoid this problem, mentions are replaced by fantasy names or just a generic label, associated in an unequivocal way to each actor in the text. This last method is used, for example, in our corpus where the labels are fictitious initials: AA, BB, etc. In our example, "Mr. Juan Pérez" and "Mr. Perez" are replaced by "Mr. AA" since both refer to the same person, while "María Rodríguez" and "Juana Fernández" are replaced by "BB" and "CC", respectively.

The process of de-identification is now more complex than just redacting: It is no longer enough to detect and suppress all names, but consistency should also be maintained in assigning the new labels to the original names. This implies a certain level of co-reference resolution, at least between the different variants of proper names. This task presents its difficulties. For example, consider the following excerpts taken from a decision of the Family Court of Appeals:

1. **Rodríguez Martínez, Juan Líber** c/ **Pérez Rodríguez, Pedro** y otros.
   *Rodríguez Martínez, Juan Líber against Pérez Rodríguez, Pedro and others.*
2. (. . . ) Sres. **Pedro y Juan Pérez**, deduce recursos de apelación.
   *(. . . ) Misters Pedro y Juan Pérez, deduct appeals.*
3. No puede considerarse que **Pedro Pérez** ha omitido contestar la demanda (. . . )
   *It can not be considered that Pedro Pérez has omitted to answer the demand (. . . )*
4. Se intimó la aceptación de **Pedro** a fs. 32 vta. y a **Juan** a fs. 36/37 (. . . )
   *It was asked the acceptance of Pedro in p. 32 and Juan in pp. 36/37 (. . . ).*

In the above example, for the substitution process, "Pérez Rodríguez, Pedro", "Pedro Pérez" and "Pedro" (fragments 1, 2, 3 and 4) refer to the same person, and "Juan Pérez" and "Juan" (fragments 2 and 4) refer to another. In particular, in fragment 2, from the

plural "Misters" and the distribution property of the conjunction *and*, it is inferred that the aforementioned "Pedro" has "Pérez" as a surname, while in fragment 4 it must be assumed that "Pedro" refers to "Pedro Pérez" because there is only one person with that given name in that particular document. The same cannot be said about "Juan", since there are two persons named as such in the document: "Juan Rodríguez" and "Juan Pérez" (fragments 1 and 2). In summary, for this example, the expected output for entities would be the following: AA:{"Rodríguez Martínez, Juan Líber"}, BB:{"Pedro Pérez", "Pedro", "Pérez Rodríguez, Pedro" }, CC:{"Juan", "Juan Pérez"}.

Different types of named entities appear in these documents: people, geographical locations, organizations. In this work, we focus on personal names. However, not all person names should be anonymized, only those specially protected such as minors or primary criminals, among others. An example of names that are not anonymized is that of the judges that make up the courts. The need for anonymization also depends on the type of case; for example, the main actors are not anonymized in a divorce proceeding, although the minors involved are.

It is worth mentioning that entities to be anonymized are seldomly mentioned in more than one document, and there is no compilation or canonical repositories available for all possible names present to work with, as, for instance, the Wikipedia for the task of linking names of renowned people.

### 4. Corpus

Our corpus consists of circa 80,000 documents, a fraction of the National Jurisprudence Base. The legal rulings measure an average of three pages (10,000 characters), and there are semi-fixed structural elements and enumeration structures, with parentheses, numerals and abbreviations of different types. The texts generally have references to other judgments or laws or articles of law. All these elements make intensive use of punctuation marks, especially the period, which proved to make more difficult the task of text segmentation.

About 17,000 documents of the corpus are manually de-identified, that is, there are about 17,000 documents for which we have both their de-identified and original versions. This gives us, a priori, a parallel corpus to work with, from which we can, for example, train a machine learning model or evaluate existing tools. Unfortunately, we find that many of these documents have failures in their de-identification process: There are names not marked as such in the text and there are labels inconsistently assigned to names. To make things worse, documents are partially rewritten during the editing process, with sections completely changed or even canceled.

In order to build a training set, the above problems render impossible the automatic extraction of samples by looking only for differences between the original and the anonymized version. Therefore, we run an algorithm to pre-tag the entities by aligning the documents, and we manually proceed to the revision of 1000 documents, identifying and labeling all the entities to be de-identified with the aid of BRAT [45]. The whole process resulted in a set of 10,102 tagged names, distributed over 997 documents (Three documents were discarded because they lacked entities to be anonymized); all labeled names correspond only to entities to be anonymized, while any other named entity was not labeled, including person names of judges, lawyers, etc.

Finally, for experimentation, we decide to take apart 797 documents as a training set, leaving the remaining 200 documents for validation. Measured in references to be anonymized, there are 7748 mentions for training and 2220 mentions for validation in those sets.

### 5. NER Training

For the sake of evaluating different off-the-shelf NER tools, we decide to run some of the state-of-the-art NLP tools over our whole corpus: CoreNLP [7], Freeling [6] and SpaCy [8].

Unfortunately, all tested tools behave poorly over our corpus, probably due to the significant difference in style and vocabulary from those documents in which the analyzers were originally trained. For example, the NER performance for SpaCy, using their best model for Spanish, is 0.3999 F1-micro over the validation set. Instead, SpaCy authors reported a 0.8986 F1-micro for the same task.

Figure 2 shows a fragment of NER modules' analyses over a document of our corpus. In this example, there are two names to anonymize—"Pérez Cabrera, María José" and "Martínez Mágico, Adolfo Ramón"—but none of the analyzers could mark them as person entities. In fact, only parts of these names are recognized, and even some of them got a wrong entity type (organization or others). None of the rest of the detected entities are completely correct.

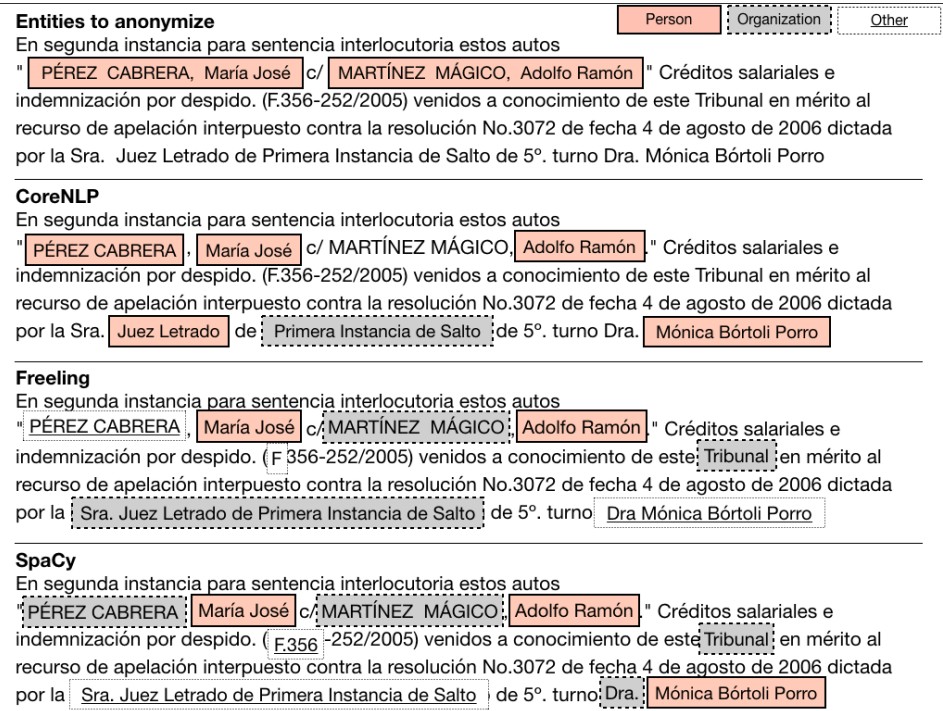

**Figure 2.** NER analysis by CoreNLP, Freeling and SpaCy over a document of our corpus.

Therefore, we decided to train an NER recognizer using our 797 manually tagged documents as the training set and the 200 documents left as the validation set. Instead of building our own tool from scratch, we decided to retrain the NER module of SpaCy. The model of SpaCy is based on a multitask CNN and allows us to load an already trained model and adjust its weights with new examples, therefore enabling us to transfer learning from existing pre-trained models. In addition, SpaCy allows per-module retrain without the need to change or re-train the rest of the modules that are before in the processing chain.

As mentioned in Section 3, not all person names should be de-identified, for example, names of judges, lawyers, etc. Since these mentions are not labeled in our corpus, the model is somehow "unlearning" to recognize certain names that should be marked as "PER" in a classical NER task, although bearing no interest for our anonymization task. A better option would be to mark even those entities, but due to resource constraints, it was not completed in time for this experiment.

All experiments started with the biggest model available for Spanish (*es_core_news_md*), which was trained over the AnCora and WikiNER corpus, and contains 20,000 unique vectors of 50 dimensions. It can perform tokenization, POS tagging, dependency parsing and NER, supporting person (PER), location (LOC), organization (ORG) and miscellaneous (MISC) entities. For the *es_core_news_md* model, the reported F1 score for PER tag is 0.8971, evaluated over the WikiNER silver corpus.

We performed different experiments, mainly changing three factors: (a) the dropout rate; (b) resetting (or not) the NER module; and (c) how we label the examples to be tagged (Table 1). We left the rest of the parameters with the default value, which corresponds to those that gave the best results for their original training set.

The drop-out rate governs the probability that individual features and internal representations are "turned off" during training; this is a technique to prevent the model from remembering the examples and avoid the over-fitting of the training set. Although this value can vary over the range (0, 1), it is usually set at 0.5 or lower [46]. We tested values ranging from 0.1 to 0.6, and, once set, the value is left static for all rounds during the same training.

SpaCy allows starting from a previously learned model or resetting the model values and starting from scratch. In the first case, we can transfer knowledge learned from a previous training (and different!) corpus and tune it to our new examples, coming from a different domain. We experimented with this feature to see how much gain could we obtain from a pre-trained model versus a NER model started from scratch.

Finally, we considered two options of labeling the entities to be anonymized in the training set: the classical tag for person, "PER", which is pre-trained in the SpaCy model, and a new subtype of this label, "PER-X", leaving the "PER" as a possible label for the NER module (although not present in our training corpus).

In Figure 3, we can see the F1-micro scores obtained with (a) label "PER-X" without resetting the NER module (PER-X), (b) label "PER" resetting (PER-0) and (c) without resetting (PER) the SpaCy NER module. "PER" experiments always achieved the best results, no matter the dropout rate selected. The best F1-micro scores are obtained for 0.35 and 0.40 dropout rates: 0.8991 and 0.8983 in 80 and 75 iterations, respectively. These values are an improvement of 0.50 points over the 0.3999 F1-micro score obtained with the model before retraining and are at the same level as the original reported 0.8986 F1-micro score. All models are trained until there is no improvement detected over the training set for 20 rounds, with a minimum training of 120 epochs.

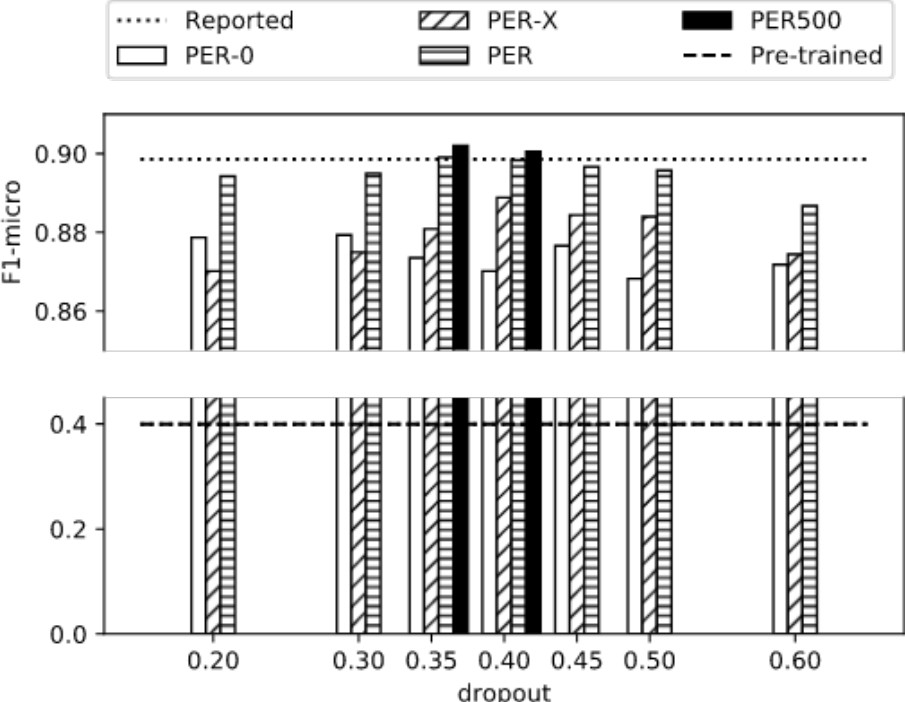

**Figure 3.** F1-micro results of training SpaCy starting from model es_core_news_md.

Following our stopping criteria, 20 rounds without performance improvement, the best results are always obtained way before 500 in all previous experiments. As a final

experiment, we trained the two best configurations for 500 epochs: "PER" label with 0.35 and 0.40 dropout rates, without resetting the NER module pre-trained values. The results obtained an F1-micro score that reaches up to 0.9021 (+0.003) and 0.9005 (+0.0022) in 420 and 185 epochs, respectively, slightly over the 0.8986 F1-micro score reported by the authors for the original model. Figure 4 shows these experiments for the 0.35 dropout rate. It can be seen that the models adjust very quickly and the gain starts to become marginal after around 100 iterations.

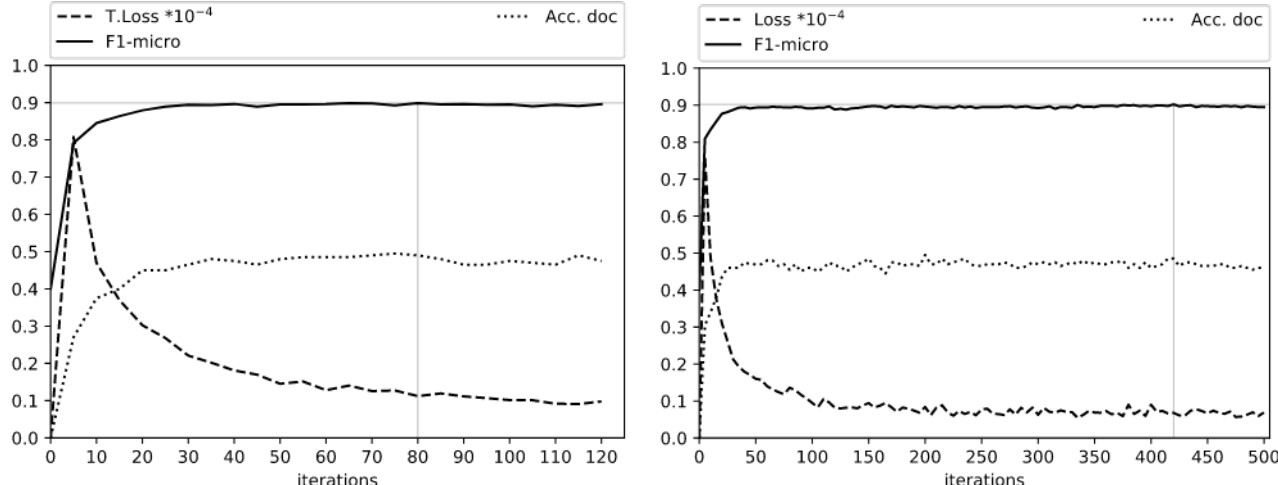

**Figure 4.** 'PER' experiment with 0.35 dropout rate for 120 and 500 epochs. Best F1-micro scores are 0.8986 in round 80 and 0.9021 in round 420, respectively.

Results for the models with the best F1-micro scores are shown in Table 2. Our best result achieves 90.21% of F1-micro, a value slightly above the 89.86% originally reported by the authors for their NER module on their test data but 50 points over the 39.99% F1-micro obtained by their model in our corpus before being retrained.

**Table 1.** Experiment parameters for NER training.

| Parameter | Value |
|---|---|
| Drop-out | 0.10, 0.20, 0.30, 0.35, 0.40, 0.50, 0.60 |
| Stop criteria | 20 rounds without performance improvement or 500 |
| NER reset | True/False |
| PER tag | Retrain PER or a new PER-X tag |

**Table 2.** Performance of the best classifier (measured by F1-micro) for each type of experiment.

| Experiment | Dropout | Iteration | F1-micro | P-micro | R-micro | Doc.Acc. |
|---|---|---|---|---|---|---|
| PER500 | 0.35 | 420 | **0.9021** | **0.8992** | 0.9050 | 0.4850 |
| PER | 0.35 | 80 | 0.8991 | 0.8792 | **0.9198** | **0.4900** |
| Reported | – | – | 0.8971 | 0.8986 | 0.8957 | – |
| PER-X | 0.40 | 205 | 0.8888 | 0.8853 | 0.8923 | 0.4500 |
| PER-0 | 0.30 | 60 | 0.8793 | 0.8515 | 0.9090 | 0.3900 |
| Pre-Trained | – | – | 0.3999 | 0.3002 | 0.5986 | 0.0000 |

Because of the the specificity of the task—only recognizing a subset of PER type, in a specific domain, in Spanish—it is difficult to compare the obtained model with previously

NER systems. For Spanish NER, several authors use CoNLL-2002 sets for evaluation: For example, in [23], an F1 score of 85.75% is reported for a CRF-LSTM model, while in [40], the F1 score goes up to 87.26% with a character+word neural network architecture. For English, training a CNN plus a fine-tuning achieved a 93.5% F1 score over the CoNLL-2003 corpus [41].

## 6. Co-Reference Resolution

As the second step in our anonymization process, entities should be replaced for labels in such a way that all references of a person receive the same (and unique) label. This is a simplified version of co-reference resolution, in which only proper names are considered while other phenomena, e.g., pronouns, are ignored, because of their lack of impact on the possibility of the re-identification of people involved.

Since we deal only with proper names, most of the entities are variations of a name—e.g., in the example of Section 3, "Pérez Rodríguez, Pedro", "Pedro Pérez" and "Pedro"—we decided to experiment with a standard unsupervised agglomerative clustering algorithm with different string distance functions to calculate the similarity of the entities detected by the NER module.

We chose the Agglomerative Clustering algorithm implemented in Scikit-Learn [47], where each document is transformed into a pre-calculated distance matrix of all its entities. The matrix is calculated by applying one particular distance to each pair of possible names.

A metric should be set to apply the Agglomerative Clustering algorithm. We took profit of distances implemented in the TextDistance project (https://pypi.org/project/textdistance/, accessed on 12 December 2021). We tested 19 distances or pseudo-distances (Overlap, Jaccard, Sorensen, Tversky, Bag, LCSSTR, Ratcliff-Obershelp, Levenshtein, Damerau-Levenshtein, Strcmp95, Jaro-Winkler, Smith-Waterman, LCSSEQ, Postfix, Needleman-Wunsch, Hamming, Goto-H, Bag, Entropy-NCD) from this package, plus a distance defined by us, based on heuristics, to determine the name and surname and compare their overlapping separately. For the sake of clarity, in this article, we report the results of our top four distances: Overlap, Jaccard, Sorensen and Bag.

Since each cluster should represent one person, and the number of mentioned persons varies from document to document, we cannot fix the global amount of clusters that we need to find. Therefore, instead of fixing the number of desired clusters, we estimate the best threshold from which two clusters should not be merged. Taking profit from our annotated corpus, different thresholds are tested for each of our chosen distances, and the ones that maximize our target performance measure over the training set are kept. The threshold values for our tests ranged from 0.05 to 0.95 with a step of 0.05 (Table 3).

**Table 3.** Experiment parameters for Agglomerative Clustering.

| Parameter | Values |
|---|---|
| Distance | Overlap, Jaccard, Sorensen, Tversky, Bag, LCSSTR, Hamming, Goto-H, Bag, Entropy-NCD , Strcmp95, Jaro-Winkler, Smith-Waterman, LCSSEQ, Postfix, Needleman-Wunsch, Ratcliff-Obershelp, Levenshtein, Damerau-Levenshtein |
| Threshold | From 0.05 to 0.95 step 0.05 |
| Linkage | Average (other options were discarded in preliminary tests) |

Different approaches can be used to calculate the distance between clusters, that is, which value the clustering algorithm will try to minimize each time clusters are merged. In our preliminary tests, the best results were systematically obtained with the average of distances between the instances of candidate clusters to be merged, and, therefore, we fix this parameter value for the rest of our experiments (Other discarded options include the maximum or the minimum distance between observations of the clusters or the variance of clusters being merged).

Names are pre-processed in order to improve results. During this process, all letters are transformed into lowercase, special accents are removed and transposed names are transformed into their canonical form. For example, the two mentions of Figure 2, "PÉREZ CABRERA, María José" and "MARTÍNEZ MÁGICO, Adolfo Ramón" are transformed into "maria jose perez cabrera" and "adolfo ramon martinez magico".

Although there are several performance metrics proposed for the general co-reference resolution problem, there is no consensus on which is the best. Among the first ones were MUC and $B^3$ [48,49], although, due to their weaknesses, several alternatives were proposed such as CEAF, BLANC and LEA [50–52]. For this task, we simply fall back to the ones normally used in clustering problems: Homogeneity, Completeness, V-measure and ARI [53], and we use the last of these mention metrics as our target metric.

In Figure 5, Completitude, Homogeneity, ARI and Document Accuracy are shown for Overlap and Jaccard by threshold value; the best ARI scores are obtained at thresholds 0.45 and 0.75, respectively, and therefore, Overlap-0.45 and Jaccard-0.75 are our best solutions for those distances. In Figure 6, the ARI performance of our top four measures variation values of distance threshold is depicted.

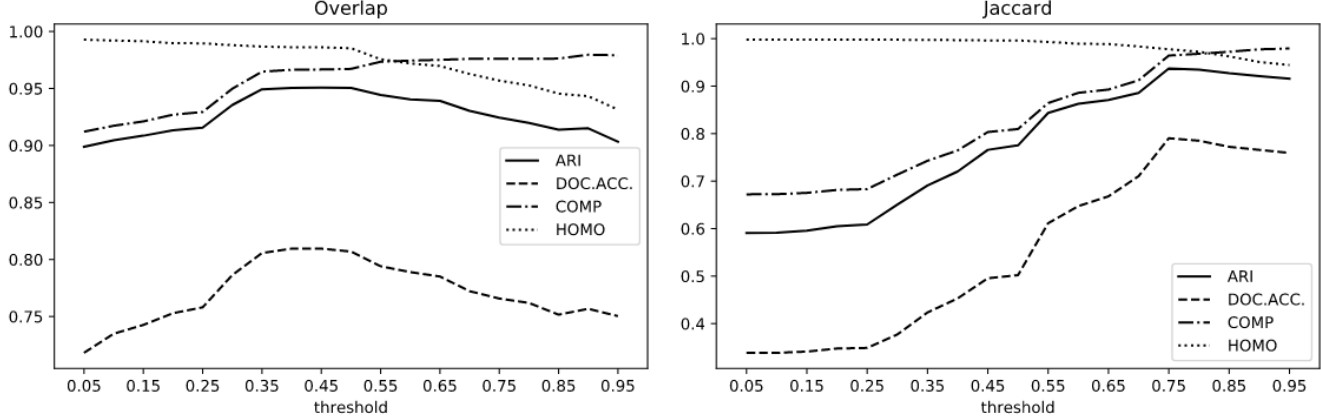

**Figure 5.** ARI, Document Accuracy, Completitude and Homogeneity for Overlap and Jaccard by threshold value.

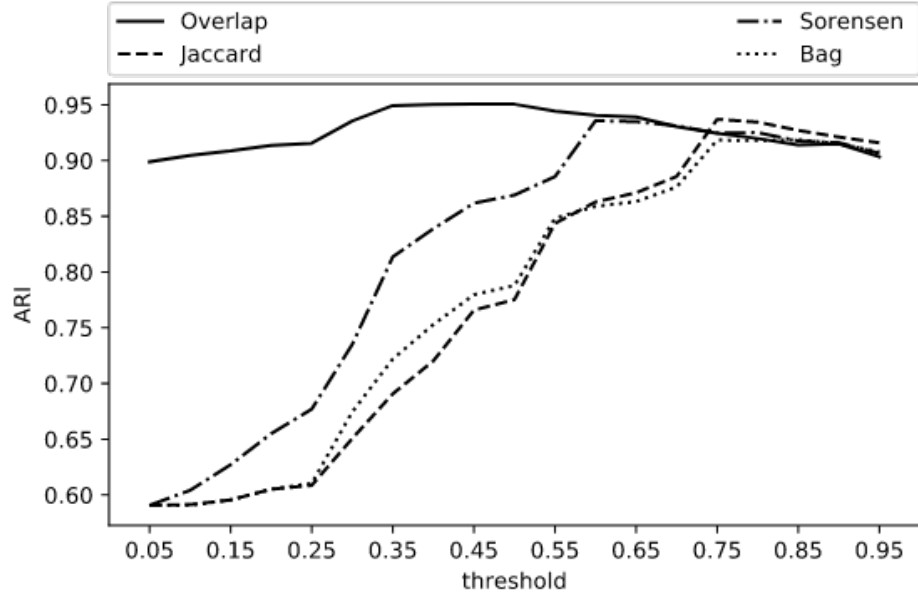

**Figure 6.** ARI for Overlap, Jaccard, Sorensen and Bag by threshold value.

After determining the best threshold for each measure, we evaluate their performance over the 2220 name references of the validation set. In Table 4 and Figure 7 the top four results, measured by ARI score, are depicted. Overlap distance with a threshold of 0.45 obtains the best results in every performance measure, with an 0.9595 ARI score over entities and an accuracy over documents of 0.81, that is, co-references are correctly and completely resolved in 162 of the 200 documents of the validation set.

**Table 4.** Performance of our four top best experiments.

| Experiment | Comp | Homo | V-Measure | ARI | Doc.Acc. |
|---|---|---|---|---|---|
| Overlap 0.45 | **0.9666** | **0.9860** | **0.9695** | **0.9506** | **0.8095** |
| Jaccard 0.75 | 0.9644 | 0.9777 | 0.9609 | 0.9370 | 0.7902 |
| Sorensen 0.60 | 0.9633 | 0.9778 | 0.9603 | 0.9357 | 0.7864 |
| Bag 0.85 | 0.9718 | 0.9528 | 0.9491 | 0.9187 | 0.7580 |

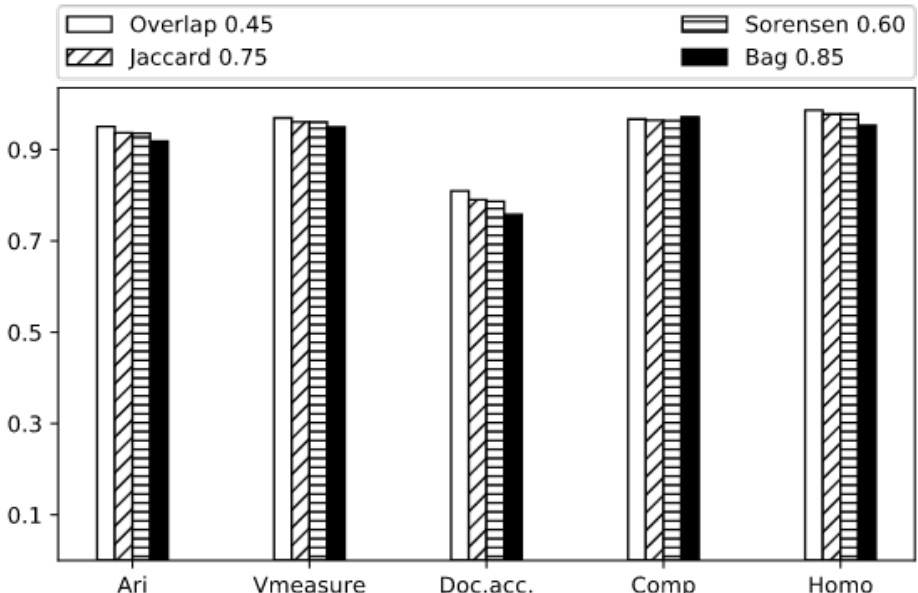

**Figure 7.** ARI, V-Measure, Document Accuracy, Completitude and Homogeneity for best Overlap, Jaccard, Sorensen and Bag experiments.

## 7. Conclusions and Further Work

This article presents a study and a prototype aiming at the automatic curation of a jurisprudence database, with a special focus on the anonymization of sensitive data. In the anonymization task, our goal is to make court rulings publicly available, protecting the privacy of people involved in the trials but without losing the coherence of the narrative of facts. Therefore, for each document, we recognize specific proper names and we solve co-references between these names to substitute them with a new unique label.

Because of the characteristics of legal texts, the performance of available NLP tools—in particular, for named entities recognition—proves the unfeasibility of applying these tools off-the-shelf without adapting them to encompass the legal domain. We experimented by retraining the NER module of SpaCy, taking profit of its pre-trained values over Ancora and Wikipedia corpus. Our best result achieves 90.21% of F1-micro, a value slightly above the 89.86% originally reported by the authors for their NER module on their test data but 50 points over the 39.99% F1-micro obtained by their model in our corpus before being fine-tuned. There is no straightforward way of comparing our work with other state-of-the-art NER experiments, due to differences in language, datasets type or even type of match

(exact match or partial match) considered. However, it is clear that our numbers are very similar to those of some of the latest reported experiments for the Spanish language.

For solving co-references, we experimented with an unsupervised agglomerative clustering algorithm, and since we do not know the number of possible clusters per document, we use our labeled corpus to estimate the best maximum distance for merging clusters. The input distance matrix is calculated using standard string functions over pre-processed names. The best results are obtained clustering with the Overlap distance, with a 0.45 merging distance threshold. This combination achieves a 95.95% ARI score and an accuracy over documents of 81%.

When we look at the results from a macro perspective, the total number of documents where all names are completely and correctly recognized and linked with each other is still below 50% of the validation set. Given the complexity of the task, these results are not unexpected and could be considered good given the number of different entities that are present in each document. Nevertheless, as future work, we plan to train our own end-to-end networks to see if we can improve the overall results of the process. We would also like to complete the performance measurements with metrics designed specifically for co-reference resolutions processes such as CEAF, BLANC or LEA.

Given the sensitivity of the data to be published, only a 100% accurate solution would allow us to turn the anonymization process into a completely automatic one. Instead, it is planned to integrate our tool as a part of an assistant that suggests options to a human operator that, in turn, validates the outcome. Since the time-consuming characteristic of the curation process, this assistant is expected to improve the output quality and also accelerate the incorporation of new court decisions documents to the NJB. In this scenery, we plan to use human changes to the suggested output as feedback to further improve the system performance.

All our experiments were conducted over a real corpus with almost 80,000 documents. There is a sanitized version of this corpus and the necessary code for the eventual reproduction of results available. In addition, as other processes such as summarization or legal arguments retrieval are likely to be developed over the same corpus, we expect that the undertaken learning processes will cooperate in this direction.

Finally, a reflection that arises is if the training and evaluation of NLP tools do not correspond, in fact, to somewhat simplified data, without enough variety to allow a more fluid passage to other application domains; it seems that obtaining good models is important, but easing the transfer of knowledge is crucial for dealing with new domains in real-world applications.

**Author Contributions:** Conceptualization, D.G.; methodology, D.G.; software, D.G.; validation, D.G. and D.W.; formal analysis, D.G.; investigation, D.G.; resources, D.G.; data curation, D.G.; writing—original draft preparation, D.G. and D.W.; writing—review and editing, D.G. and D.W.; visualization, D.G.; supervision, D.W.; project administration,D.G. and D.W.; funding acquisition, D.G. and D.W. All authors have read and agreed to the published version of the manuscript.

**Funding:** This research was funded by the National Agency for Research and Innovation (ANII) and the Ministry of Industry and Energy of Uruguay (MIEM), grant FSDA-143138.

**Institutional Review Board Statement:** Not applicable.

**Informed Consent Statement:** Not applicable.

**Data Availability Statement:** The data that support the findings of this study are available on request from the corresponding author, D.G. The data are not publicly available due to information that could compromise the privacy of participants.

**Acknowledgments:** We are grateful for the contribution of the staff of the Judicial Branch, in the areas of Jurisprudence and Technology. In particular, we would like to mention María Luisa Tosi, Gustavo Beiro, Horacio Vico and Jorge Gastaldi.

**Conflicts of Interest:** The authors declare no conflict of interest.

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
