# Peer review of "Automatic Curation of Court Documents: Anonymizing Personal Data"

_information, doi:10.3390/info13010027_

Round 1
Reviewer 1 Report
The paper focuses on a well-known (though still open) problem. The approach is quite realistic and the prospect of using the tool as an aid to humans is acceptable. Working together humans and machines should allow further progress.
I have just one suggestion about lexicon: to replace "sentence" with "opinion" all the paper along. In legal lexicon the decision of a case is "decision", "opinion", "ruling"..., while "sentence" is the quantity of time the guilty person has to spend in jail (e.g. he was sentenced three years). I suggest using "opinion" because the authors worked on the explanations given by judges about their "decisions".
Author Response
First of all, we would like to thank the reviewers for reading our
paper and take their time to make all the corrections and send us
their very welcome feedback.
Reviewer 1:
- Replace "sentence" with "opinion" all the paper along
We replaced sentence by opinion, ruling, etc.
Reviewer 2 Report
This paper describes work examining anonymizing personal data within court documents. This goal is potentially extremely valuable and, if done well, it could offer valuable insights towards the automation of Data Protection tasks in common law. The outline of the study is as follows: Introduction, De-identification of legal texts, Corpus, NER Training, Co-reference Resolution and Conclusions and Further Work. The authors experimented by retraining the NER module of SpaCy and utilizing an unsupervised agglomerative clustering algorithm. The authors conclude that, given the low success rate, their process can be utilized as “an assistant that suggests options to an operator that validates the outcome”
The paper could be better organized by clearly separating “Related Work” (literature overview) from the Introduction. While the authors provide numerous references and additional information to existing literature, relevant literature presentation is rather short. The reader is left puzzled about the set of related problems that cover the domain of the problem in question, what has been already proposed and so on. Additionally, I would expect the authors to clearly highlight the differences between their approach and what has already been done.
Furthermore, while numerous references study anonymization techniques in medical records/ health data, fewer referenced studies deal with legal documents. The authors should enhance the corresponding section with studies from the legal domain.
The paper is readable and the research goal is clear. However, the main drawback concerns the novelty. While relevant prior literature is referenced the authors omit to present their previous paper(s) (e.g. “Towards De-identification of Legal Texts” - arXiv:1910.0373). The authors should enhance relevant literature, clearly explaining the differences between previous work(s) and the current one.
Authors state that they “present a project aimed at the automatic curation of our National Jurisprudence Database”. It is my understanding that the authors actually present a study and not a project. I would advise authors to substitute project with study in various places within the paper.
Figure 2 and the corresponding passage, describe a legal citation graph which is clearly out of the scope of the current study. Please omit unnecessary and unrelated material from the study.
While the study examines NER Training results and also Co-reference Resolution results I am puzzled on the overall success rate of the pipelined methods. Have the authors tested the effectiveness of their method as a whole and if so what where their findings?
Finally, in the conclusion, I would expect clear ideas of future work on how to overcome observed limitations and addressing identified errors
Author Response
First of all, we would like to thank the reviewers for reading our
paper and take their time to make all the corrections and send us
their very welcome feedback.
Reviewer 2:
- The paper could be better organized by clearly separating “Related
Work” (literature overview) from the Introduction.
The introduction was reorganized and a "related work" section was
added for clarity.
- The authors should enhance the corresponding section with studies
from the legal domain.
We added some reference works that deal with anonymization for the legal domain.
- The authors omit to present their previous paper(s) (e.g. “Towards
De-identification of Legal Texts” - arXiv:1910.0373)
This paper is a pre-print of this work and it was never published in
any conference proceedings or any other type of publication.
- I would advise authors to substitute project with study in various
places within the paper.
We address this issue and substitute project by work, study, etc.
- Please omit unnecessary and unrelated material from the study.
The detection of reference and the associate graph was another part of
the original study, but we understand that it might seem out of focus
in this paper. As suggested, we removed it.
- I am puzzled on the overall success rate of the pipelined methods.
Have the authors tested the effectiveness of their method as a whole
and if so what where their findings?
We made a macro evaluation (is the whole document correct or not?) and
reported it has a 50% accuracy (although migh seem low, it was
expected). We would like to add more metrics in order to measure
differently the complete chain of entities, but wef left (for the
moment) as a future work.
- in the conclusion, I would expect clear ideas of future work on how
to overcome observed limitations and addressing identified errors
We modify the conclusions and we expect our future work is now
clearlier explained.
Round 2
Reviewer 2 Report
This paper describes work examining anonymizing personal data within court documents. This goal is potentially extremely valuable and, if done well, it could offer valuable insights towards the automation of Data Protection tasks in common law. The outline of the study is as follows: Introduction, Related Work, De-identification of legal texts, Corpus, NER Training, Co-reference Resolution and Conclusions and Further Work. The authors experimented by retraining the NER module of SpaCy and utilizing an unsupervised agglomerative clustering algorithm. The authors conclude that, given the low success rate, their process can be utilized as “an assistant that suggests options to an operator that validates the outcome”
Presentation of the experiments and results obtained is not adequate. The authors should present all the tested approaches, the metrics utilized, justify upon the suitability of the chosen metrics, and present results in a uniform fashion.
1) Firstly, the authors should properly justify in the paper the use of ad-hoc values, e.g., line 301 ranging from 0.1 to 0.6, line 320 As a final experiment, we trained the two-best configuration for 500 epochs and so on ... , that reduces the generalizability of the proposed approach. The authors should also properly justify/explain how they were selected and if they affect the results, and how?
2) Lines 371-373: “In order to calculate the distance between clusters, the algorithm is configured to the average of distances between members of each of the clusters involved”.
Please explain why you choose “average of distances” instead of other methods?
3) The author state in lines 361-362 “More than ten distances or pseudo-distances are tested, Overlap, Jaccard, Sorensen, Bag 362 and Hamming are in between them.”
Please explain how many distances are tested and state them accordingly.
Regarding the Experimental Setup, I would also strongly encourage the authors to make the reader's life easier by providing a table summarizing testing parameters and their corresponding ranges
While the study examines NER Training results and also Co-reference Resolution results I am puzzled on the overall success rate of the pipelined methods.
Have the authors tested the effectiveness of their method as a whole and if so what where their findings? Please provide your methodology, metrics, results and so on.
Finally, given the plethora of similar/related algorithms in the literature, I do believe that the authors should properly justify in the paper their choice of methods. The reader is left with the impression that the authors choose those algorithms simply because they had access to the source code/ or toolkits to utilize them.
Also highlighting statistically significant values between the tested methods would offer readers a more convincing and detailed comparison of the tested methods.
Author Response
Thanks for the review, again.
Following modifications were performed:
- We describe NER and Clustering parameters involved in our experiments
- We state and values that were tested
- Added tables with parameters involved
Regarding the whole evaluation (end-to-end pipeline), we would like to add more metrics in order to measure differently the complete chain of entities, but it was left as a future work.
Best regards
